# What is the overlap between malnutrition, frailty and sarcopenia in the older population? Study protocol for cross-sectional study using UK Biobank

Nada AlMohaisen[1,2,3], Matthew Gittins[1,3], Chris Todd[1,3,4], Sorrel Burden[1,3,5]*

1 School of Health Sciences, University of Manchester, Manchester, United Kingdom, 2 Department of Clinical Nutrition, College of Applied Medical Sciences, Imam Abdulrahman Bin Faisal University, Dammam, Saudi Arabia, 3 Manchester Academic Health Science Centre, Manchester, United Kingdom, 4 Manchester University Foundation NHS Trust, Manchester, United Kingdom, 5 Salford Royal Foundation NHS Trust, Salford, United Kingdom

* sorrel.burden@manchester.ac.uk

**Data Availability Statement:** All this study protocol files are available from the UK Biobank database (accession URL: https://biobank.ndph.ox.ac.uk/showcase/).

## Abstract

### Background

In an increasingly older adult population, understanding the inter-relationship between three age related conditions malnutrition, frailty and sarcopenia is important in order to improve their recognition, treatment and prevention. This study aims to determine the overlap between malnutrition, frailty and sarcopenia by measuring estimates of prevalence for each individual condition. In addition, we will compare two models of frailty which are the accumulation of deficits and phenotype models.

### Methods/design

This is a cross-sectional study that will use the UK Biobank database, which will include a subset of 381,000 participants: males and females aged 50 years and above who completed the baseline assessments. For the baseline assessments, details of the participants' characteristics will be included. All three conditions will be identified and mapped to variables collected at the baseline assessment. Variables for malnutrition will be mapped according to the Global Leadership Initiative on Malnutrition (GLIM) criteria. Frailty will be defined according to two models: the 36 deficits and the phenotype model. Finally, sarcopenia will be assessed according to the European Working Group on Sarcopenia in Older People (EWGSOP) standard.

### Discussion

This proposed study will help to understand the presence of malnutrition, frailty and sarcopenia in the older population and describe any overlap between the conditions. There is little published research on the overlap between these three conditions. Despite the similarity and shared criteria used for the identification of malnutrition, frailty and sarcopenia there is

**Funding:** Yes, this PhD is scholarship from the Saudi Arabia cultural bureau in London, which they pay the tutors without interfering with the works. The funders had no role in study design, data collection and analysis, decision to publish, or preparation of the manuscript.

**Competing interests:** NO authors have competing interests.

**Abbreviations:** BIA, Bioelectrical impedance; BMI, Body Mass Index; ELSA, English Longitudinal Study of Ageing; EWGSOP, European Working Group on Sarcopenia in Older People; FI, Frailty Index; GLIM, Global Leadership Initiative on Malnutrition; HCS, Hertfordshire Cohort Study (UK); ICD-10, International Statistical Classification of Disease and Related Health Problems, 10th Revision; LTCs, long-term conditions; MET, Total Metabolic Equivalent; MSS, Malnutrition-Sarcopenia Syndrome; MUST, Malnutrition Universal Screening Tool; NHANES III, Third National Health and Nutrition Examination Survey (USA); NRES, National Research Ethics Service; WHO, World Health Organization.

still a lack of cohesive thinking around the overlap of applied definitions and identification criteria.

## Trial registration

ClinicalTrials.gov NCT04655456 approved on the 10th of December 2020.

## Background

The population around the world is ageing rapidly. It is expected that by 2050, the number of people older than 65 years worldwide will increase to two billion [1]. In England, the number of older people reached 18% of the population in 2016 [2]. Regardless of the decline in fertility and birth rates, all countries are experiencing an increase in life expectancy [3]. This increase has resulted in a rising demand for, and pressure on, local services. There is a particular demand for these services within the National Health Service (NHS) in the UK, where budget spending on older people grew from two percent to 16% between 2007 and 2016 [4]. Therefore, the health status of older members of the population is becoming a priority. A healthy dietary intake and active lifestyle are key components for healthy aging and are well recognised as factors associated with a lower risk of developing lifestyle-related diseases [5–7].

The aging process involves changes in body functions including alterations in the sense of smell and taste and capacity to chew and swallow [8]. These changes directly affect appetite and food intake [8], which can have an impact on nutritional status and can lead to malnutrition in the form of undernutrition [8]. There are two forms of malnutrition undernutrition, and being overweight or obese which results from diet-related non-communicable diseases [9, 10]. Cachexia is a another nutritional disorder which can be define as "a complex metabolic syndrome associated with underlying illness and characterized by loss of muscle with or without loss of fat mass" [11]. Malnutrition "undernutrition" is common in older people and generally increases with aging [12, 13]. It is crucial to detect malnutrition in older people because it causes a decline in overall health [14]. It can worsen the deterioration of health and functional status, and increase the risk of dependency and disability [12]. It is associated with undesirable health conditions, including an increased likelihood of morbidity and mortality [15, 16], an increased risk of falls [17], reduced cognitive function, and decreased bone mass in older people living in community dwellings [18, 19]. Furthermore, older malnourished inpatients have a higher number of complications and longer hospital episodes, which result in increased associated healthcare costs [10].

Over the last few decades, many criteria have been proposed for identifying malnutrition and many tools have been implemented. Most of these tools include a combination of the same variables or criteria: weight loss, body mass index (BMI) and signs of eating difficulties [20]. One of the most recently proposed definitions of malnutrition emerged from the Global Leadership Initiative on Malnutrition (GLIM). Here malnutrition is determined using a phenotype and aetiological model, which includes three phenotypes (non-volitional weight loss, low BMI and reduced muscle mass) and two aetiological criteria (reduced food intake and inflammation) [21]. The GLIM has broader criteria because it was developed using previous screening and assessment tools to ensure it is applicable in all countries and transferable to all settings. By using the GLIM criteria malnutrition can be compared in different settings through prevalence, incidence, intervention and outcome [21].

In addition to malnutrition, two other conditions frailty and sarcopenia are widespread in older populations. All three conditions are characterised with a loss of body tissue, which can result in negative outcomes such as an increased risk of falls [22], poor quality of life [23], increased risk of mortality [24, 25]. Malnutrition and frailty both have other similar characteristics in the assessments used, including, functional and cognitive status. Thus there is overlap in the definition and treatment of these conditions [26].

Frailty is associated with ageing and may lead to premature death [27, 28], multi-morbidity, specific long-term conditions and mortality [29, 30]. In addition, frailty has been shown to decrease the quality of life of older people [31]. It can be defined as a 'state of increased vulnerability to poor resolution of homoeostasis after a stressor event' [32]. Two of the most widely used models used to identify frailty are the phenotype model, which describes a group of patient characteristics [32], and the cumulative deficit model, which assumes an accumulation of deficits that happen with ageing [33, 34].

The phenotype model describes frailty as a 'clinical syndrome in which three or more of the following criteria are present: unintentional weight loss (10 lbs in the past year), self-reported exhaustion, weakness (grip strength), slow walking speed and low physical activity' [35]. This model is used as the definition to diagnose frailty in many studies [36].

The cumulative deficit model was proposed by Rockwood and Mitnitski and was validated in the Canadian study of Health and Aging [36, 37]. This model used an index that considers symptoms, signs, diseases and disabilities as deficits that are combined into an FI to determine frailty [38]. The original cumulative deficit model was based on 92 variables [39]. Subsequent research revealed that it could be reduced to 30 variables, without a loss in predictive validity [40]. The cumulative deficit model calculates the FI by defining the numbers of health deficits present in a person, and then dividing by the number of health deficits considered [38, 41].

In addition to malnutrition and frailty, the older adult population may also experience the related condition sarcopenia. In 2000, the World Health Organisation(WHO) estimated that 600 million people aged 60 or older worldwide were diagnosed with sarcopenia [42]. This number is predicted to increase to 1.2 billion by 2025, and to double by the year 2050 [42]. A study suggested that frailty and its relation to sarcopenia, malnutrition and other factors is a network called the 'geriatric web of frailty' [43]. It argued that frailty is a complex condition caused by many interacting factors, such as falls, fractures, impaired protein synthesis, malnutrition and sarcopenia [43]. Many researchers have shown that the overlap between frailty and sarcopenia is due to the shared features between them, such as weight loss and low physical capability. This is because both conditions involve a failure in skeletal muscle (low physical function) that is detected through grip strength and gait speed [35, 44].

People with sarcopenia have specific characteristics, including a low body weight and a deterioration in physical performance [45]. It can be defined as 'a progressive and generalised skeletal muscle disorder that is associated with an increased likelihood of adverse outcomes including falls, fractures, physical disability and mortality' [46]. Older adults with sarcopenia have reported poorer general health and physical function scores compared to participants without sarcopenia [45].

The European Working Group on Sarcopenia in Older People (EWGSOP) recommended criteria for diagnosing sarcopenia. These include low muscle mass, low muscle strength and lower physical performance [46]. In order to detect sarcopenia, two parameters are needed pertaining to the amount of muscle and its function. These can be translated into measurable variables: muscle mass, muscle strength, and physical performance [47]. To measure muscle mass, there are many techniques that could be used for assessment. These include computed tomography, magnetic resonance imaging, dual energy X-ray, and

bioimpedance analysis [47]. The measurement of muscle strength has fewer validated techniques, and one of the most widely used is handgrip strength because it is easy to use and readily available [48]. Others include knee flexion or extension or peak expiratory flow. There are a number of methods that have been used for physical performance, including the short physical performance battery, gait speed, the timed get-up-and-go test and the stair climb power test [47]. However, the gait speed test is usually used for clinical practice and research [47].

There are different cut-off points when defining sarcopenia that are based on the measurement technique used and the availability of reference standards. The EWGSOP recommends the use of a reference value derived by calculating two standard deviations below the mean from data collected from healthy, young adults rather than other predictive reference population cut-off points [47].

The aim of this study is to explore the overlap between malnutrition, frailty and sarcopenia in an older population, using UK biobank data, by determining the estimated prevalence of these three conditions.

## Methods/design of UK Biobank

This is a cross-sectional study using participants recruited to the UK Biobank. The UK Biobank is a population-based prospective longitudinal cohort with information on approximately 500,000 people. Participants were recruited from the NHS aged between 40–69 years. However, the database included participants aged 37 to 73 for the baseline visit. Data were collected for the initial assessment visit between 2007 and 2010. This was termed the baseline assessment that will be used for this study. An invitation was sent by mail to people who were living within a reasonable travelling distance from the assessment centres used for data collection. People were recruited from England, Scotland and Wales.

At the baseline assessment visit, all participants had to be able to give consent before starting the assessment. Then a series of assessment stations where undertaken that included a touchscreen questionnaire and an interview questionnaire about diet, cognitive function, work history and digestive health [49]. In addition to demographic data, the participants were given cognitive function tests, their height, hip and waist measurements were recorded, as well as their bioelectrical impedance measurements and hand-grip strength. The UK Biobank database aims to improve prevention, diagnosis and treatment of a broad number of diseases. This detailed information on participants provides a resource for investigators to conduct research relating to particular diseases [49].

The UK Biobank Ethics and Governance Council (EGC) was founded by the Medical Research Council and Wellcome Trust, allowing them to act independently [50]. The UK Biobank controls all access to and use of the data [50]. However, UK Biobank evaluates all proposals to verify that they would be compliant with the participants' consent and the Biobank's Governance Framework and have received the required ethics approval [50]. The researcher and collaborators signed a Material Transfer Agreement (MTA) with the UK Biobank. The MTA was submitted and the payment process completed for the approved application, allowing the data to be released to the approved researcher.

### Eligibility and exclusion criteria

The participants included in the study will be both male and female, aged 50 years or older, who completed a touchscreen questionnaire, and physical measurements. Any participant who is under 50 will be excluded.

## Study sample size

This study will be a subset of the main sample size from the UK Biobank, and it is anticipated 381,000 participants will meet the inclusion criteria (76% of the total sample of participants in the UK Biobank, who met the inclusion criteria age 50 and above) [29].

## Baseline characteristics

Participant characteristics that will be recorded are detailed in Table 1.

At the baseline assessment participants' medical histories were recorded including a list of comorbidities that is described as 43 long-term conditions similar to the Hanlon study [29]. In addition, we will include other characteristics as shown in Table 1.

**Mapping malnutrition to biobank variables.** The GLIM criteria will be applied to variables collected in the UK Biobank database in order to define participants with malnutrition. It includes five criteria, which will be mapped to equivalent variables that are available in the UK Biobank shown in Table 2.

These variables will represent three phenotypic criteria and two etiologic criteria, where patients will be defined as having malnutrition if they have one phenotypic criterion and one aetiological criterion. The three phenotypic criteria are BMI $< 22$ kg/m$^2$ if participants are 70

**Table 1. Baseline participant characteristics.**

| Variable | Data Description | Categories Description/Units of measurement |
|---|---|---|
| Age | reported in years | |
| Gender | male or female | |
| Smoking status | prefer not to answer, never, previous and current. | |
| Medications use | number of medications (treatments) were entered by participants in the questionnaire. | |
| Alcohol intake | Seven categories: daily, almost daily, three or four times a week, once or twice a week, one to three times a month, special occasions only, never and prefer not to answer. | |
| Education level | prefer not to answer, none of the above and other professional qualifications. | Other professional grouped: NVQ, HND, HNC or equivalent; CSES or equivalent; an O levels/GCSEs; an A levels/AS levels or equivalent; or a college or university degree |
| Occupation | Ten groups: manager and senior officials, professional occupations, associate professional and technical occupations, administrative and secretarial occupations, skilled trade occupations, personal service occupations, sales and customer service occupations, process plant and machine operatives, elementary occupations or other jobs | |
| Income | Seven groups: less than 18,000; 18,000 to 30,999; 31,000 to 51,999; 52,000 to 100,000; greater than 100,000; and the 'do not know' and the 'prefer not to answer' groups | Pounds sterling |
| Ethnicity | Eight categories: White, Mixed, Asian or Asian British, Black or Black British or Chinese; and another ethnic group, do not know, and prefer not to answer. | |

NVQ = National Vocational Qualification, HND = Higher National Diploma, HNC = Higher National Certificate, CSES = Certificate of Secondary Education, GCSEs = General Certificate of Secondary Education or equivalent

**Table 2. Variable mapped to malnutrition, frailty and sarcopenia.**

| Variable | Data Description | Categories Description/Units of measurement | Malnutrition | Frailty | Sarcopenia |
|---|---|---|---|---|---|
| Dietary intake | FFQ and 24-hour dietary recall | | X | | |
| Anthropometric measurements | height measured using a Seca 202 height measuring rod and weight measured using Tanita BC-418 MA body composition analyser | Height recorded in centimetre unit Weight recorded in kilograms unit | X | X | X |
| Body composition | Bioelectrical impedance using Tanita BC-418MA body composition analyser | used to determine body composition fat mass, fat-free mass and body water. | X | X | X |
| Handgrip strength | Jamar J00105 hydraulic hand dynamometer | Kilogram | | X | X |

years or older, or BMI $< 20$ kg/m$^2$ if participants are under 70 years old [21]; weight loss, as reported by the participants' response to 'Compared with one year ago, has your weight changed?'; and finally fat free mass index, calculated by dividing fat free mass by height squared (FFMI, kg/m2). The thresholds will be $<17$ kg/m$^2$ for males and $< 15$ kg/m$^2$ for females [20].

The first etiologic criteria, is defined as reduced food intake $< 50\%$ reduction of the standard energy requirement for the male group (50–54 years: 2,581 kcal/day, 55–64 years: 2,581 kcal/day, 65–74 years: 2,342 kcal/day) and the female group (50–54 years: 2,103 kcal/day, 55–64 years: 2,079 kcal/day, 65–74 years: 1,912 kcal/day) [51]. The second etiologic criterion is inflammation based on the C-reactive protein (CRP). [11, 52].

**Mapping frailty to biobank variables.** Two models will be used to define frailty and compared. Only one model will be used to consider the overlap with malnutrition and sarcopenia. The 36 deficits model, has been constructed and validated, and applied to large data sets previously [39]. It will be matched to equivalent variables recorded in the UK Biobank. These are listed in S1 Table. The cumulative deficit model calculates the FI by defining the numbers of health deficits present in a person and then dividing by the number of health deficits considered [38, 41]. For instance, a participant with three deficits would have an index score of 3, which out of 36 equals a 0.083 FI. The threshold levels will be: low (robust), where the FI is less than or equal to 0.2; medium (pre-frail), where the FI is more than 0.2 and less than or equal to 0.35; and high (frail), where the FI is more than 0.35. These levels were suggested by Kulminski and colleagues and used in studies that looked at frailty, bone health, and overall mortality [53, 54].

The phenotype model, includes five criteria that will also be mapped with available variables in the UK Biobank. The first four criteria are derived from the self-reported variables entered by participants including: weight loss, exhaustion, slow walking pace and low physical activity. Hand grip strength is a measured variable See Table 2. These criteria were previously used in two studies based on the UK Biobank database [29, 55]. Participants will be classified as robust (met none of the frailty criteria), pre-frail (met one or two of the criteria) or frail (met three or more of the criteria) according to cut-offs described by Fried and colleagues [35]. This is done to ensure consistency with the existing literature, refer to Table 3.

**Mapping sarcopenia.** The EWGSOP standard definition of Sarcopenia has three criteria that can be mapped to variables in the biobank. Sarcopenia is diagnosed if the person has low muscle mass in addition to criterion two or three. The first is muscle mass measured by bioelectrical impedance and the second is muscle strength, measured by handgrip strength (see Table 2). Finally, physical performance as assessed by the participants' self-reported answers to the validated International Physical Activity Questionnaire [56].

**Table 3. Mapping the phenotype model.**

| Variable | Data Description | Categories Description/Units |
|---|---|---|
| weight loss | The information is provided by the participants' answer to the question 'compared with one year ago, has your weight changed?' | Response: yes, lost weight = 1, other = 0 |
| exhaustion | Determined by the question 'over the past two weeks, how often have you felt tired or had little energy?' | Response: more than half the days or nearly every day = 1, other = 0 |
| walking speed | Determined from answers to the question 'how would you describe your usual walking pace? | (1) slow pace (2) steady/average pace or (3) brisk pace |
| physical activity | Derived by using both the Total Metabolic Equivalent [MET] and the physical activity questionnaire | |
| handgrip strength | Cut-points classified according to gender | men less than 30kg women less than 20kg |

## Statistical analysis

All data analysis was performed using Stata V16/MP (Stata Corporation, College Station, TX, USA).

To describe the overlap of the three conditions, a Venn diagram will be produced. Due to the lack of independence as the conditions overlap prevalence estimates and the corresponding confidence intervals will be calculated for each segment.

Estimates of prevalence for malnutrition, frailty and sarcopenia will be calculated for the number of older people included at baseline in the study. The number identified with each condition of interest (numerator) will be divided by the total sample (denominator). The resulting proportion will be multiplied by 100,000 to give the number with each condition per 100,000 persons aged over 50 years old in the population. This 'crude' estimate is likely to be biased due to differences between the age and gender structure of the Biobank and the UK population. To provide a more representative estimate of prevalence, a direct standardisation approach will be applied. The Biobank sample will be stratified by age groups (e.g., 50–54, 55–59, 60–64, etc.) and gender (male/female), and the prevalence within each group will be estimated. These will then be weighted (using direct standardisation) based on the age and gender population of the UK. This comparison population for the UK will be obtained from the Office of National Statistics for the equivalent time period (2008). If possible, we will repeat the direct standardisation adjustment for ethnicity, socioeconomic status and education level. In each case, 95% confidence intervals will be calculated for the population proportion.

To compare the agreement between the two methods used to measure frailty, we will describe and compare the proportion estimate based on the overlap in 95% confidence intervals and the level of agreement with a Kappa comparison. The Kappa comparison indicates the level of agreement compared to random chance alone. In this comparison, a value of one means a perfect agreement compared to zero, which indicates a level of agreement no better than random chance alone. For intermediate values, Landis and Koch (1977) suggested the following interpretations: below 0.0: poor; 0.00–0.20: slight; 0.21–0.40: fair; 0.41–0.60: moderate; 0.61–0.80: substantial; 0.81–1.00: almost perfect.

From the baseline assessment, details of the participants' characteristics anthropometric measurements, participants' medical histories, smoking status, uses of medications, dietary intake records and alcohol intake will be used. This descriptive information will be considered covariates. In this study any missing data present in the key variables related to determining the outcome, i.e., the presence of the condition, as well as the key descriptive covariates, will be described and details will be given on how the variables influence the results.

## Discussion

Research describing the three conditions- malnutrition, frailty and sarcopenia- and their overlap in the population is currently limited. Few researchers have studied the effect of two of the three conditions in older adults [26, 57], but the three conditions are rarely studied together [43]. In a group of 92 selected nursing home residents aged 75 and above all three conditions were present in 7%. Twenty-nine (32%) residents were neither malnourished, sarcopenic nor frail. In addition, 5% of the residents were malnourished and simultaneously sarcopenic and frailty coexisted with sarcopenia in 10% of the subjects. In a group of 92 residents, selected from nursing home aged 75 and above, were measured for the overlap between malnutrition, frailty and sarcopenia. It showed all three conditions were present in 7% of residents, while 32% of the participants were free from malnutrition, frailty, and sarcopenia. In addition, 5% of residents were malnourished with sarcopenia in the nursing home and frailty coexisted with sarcopenia in 10% of subjects [58]. These three conditions affect mobility and increase the risk of disability, which has been shown to shorten lifespans [29, 45, 59]. One study reported an indicator called healthy life years (HLY) and measured it across 14 European countries between 1995 and 2003. The results showed that HLY indicator was stagnated in UK males, and decreased in UK females aged 65 years and above [60, 61]. This highlights that early identification of these conditions in older adults is vital so treatment and interventions can be initiated in this age group [62]. However, the evidence behind these conditions is still developing and an evidence gap still exists. Thus, further studies are needed in order to describe then understand the overlap between malnutrition, frailty and sarcopenia.

This proposed research aims to investigate the estimated prevalence of three conditions—malnutrition, frailty and sarcopenia—in people aged 50 years and above who are living in community dwellings using the UK Biobank database. In order to conduct this study, many variables need to be mapped with the available variables in the UK Biobank database. It will be challenging, as not all variables in the diagnostic criteria for malnutrition, frailty and sarcopenia are in the database. For example, the GLIM criteria will be used to identify malnourished older people within the UK Biobank participants. In the phenotypic criteria, we will map weight loss using the participants' answers to the question 'compared with one year ago, has your weight changed?', and not by using the amount of weight recorded. For frailty, the 36 deficits will be matched with the UK Biobank database. However, not all 36 deficits variables were available in the database. For example, the deficit 'best mobility and transfer problems' was mapped with an attendance/disability/mobility allowance variable in the database, which has the best-matched information. Although we have made as much effort as possible to match equivalent variables there will be some variables where matches are not as specific as they could be. However, the sarcopenia EWGSOP diagnostic criteria was matched with similar variables in the database with no issue. In addition to these factors the biobank data is not representative of the UK population as a whole [49]. However, the findings of this study will be standardised using data from a second source so the results will be more generalisable.

Other studies have used biobank data and published results mapping definitions of sarcopenia and frailty to available biobank variables [29, 55, 63] and these studies have been considered in this study design.

### Study limitations

This study has several limitations. First, the study uses a secondary database (the UK Biobank) not designed specifically to answer this research question. This comes with some limitation because there is a degree of selection bias in the sample. The UK biobank has a healthier population compared to the general population in the UK thus, it is expected that the prevalence of

the three conditions may be lower [64]. Secondly, this study is dependent on the available variables (handgrip, BMI, physical activity, walking pace, exhaustion) to detect the prevalence of frailty, although these measures are surrogate indicators for frailty. In addition, some variables used for the identification of malnutrition, frailty and sarcopenia were self-reported by the participants. Standard operating procedures were used for measurements at the baseline visits for anthropometry, however it is unknown if the equipment was calibrated. Lastly, the EWGSOP criteria was used for identify participants with sarcopenia. however, this has now been updated (EWGSOP2). There are two differences in the updated version, which are the use of muscle strength as a criterion rather than low muscle mass [46]. And low physical activity has been excluded [46]. There have been a number of studies comparing EWGSOP1 and EWGSOP2, which showed good levels agreement [65–68]. Other studies, indicate a lower prevalence of sarcopenia when using EWGSOP2 [69]. Therefore, this study plans to use EWGSOP1 for identifying participants with sarcopenia. However, the biobank provides a unique opportunity to look at malnutrition, frailty and sarcopenia in an older population due to the physical measurements recorded in conjunction with dietary and nutritional variables.

## Summary

This study will include a very large sample, which will strengthen both internal and external validity. Furthermore, the variables in the Biobank have been measured by trained researchers and documented in detail. The proposed study is also strengthened by its novelty: as it is one of the first studies to investigate the overlap between malnutrition, frailty and sarcopenia in the elderly population living in the UK.

## Declarations

### Ethical considerations/ ethics approval and consent to participate

Anonymised data will be downloaded from the Biobank database. Thus, the researcher will not be required to obtain ethical approval or gain consent from participants, because it is secondary analysis. However, the UK Biobank already has ethical approval and all UK Biobank participants' signed consent forms and must be registered with general practitioner. They stated that 'it has obtained Research Tissue Bank (RTB) approval from its governing Research Ethics Committee (REC), as recommended by the National Research Ethics Service (NRES)' [70]. Nevertheless, precautions will be considered in dealing with the data as mentioned previously in data management section, to ensure data is handled appropriately.

## Supporting information

**S1 Table. 34-cumulative deficits matched to variables from UK Biobank.**
(DOCX)

**S1 Checklist. Checklist of items that should be included in reports of observational studies.**
(DOCX)

## Acknowledgments

I would express my deep gratitude to the participants who gave their time and consent for participating in the UK Biobank database.

## Author Contributions

**Conceptualization:** Nada AlMohaisen.

**Data curation:** Nada AlMohaisen.

**Formal analysis:** Nada AlMohaisen.

**Funding acquisition:** Nada AlMohaisen.

**Investigation:** Nada AlMohaisen.

**Methodology:** Nada AlMohaisen, Matthew Gittins.

**Project administration:** Nada AlMohaisen.

**Resources:** Nada AlMohaisen.

**Software:** Nada AlMohaisen.

**Supervision:** Nada AlMohaisen, Matthew Gittins, Chris Todd, Sorrel Burden.

**Validation:** Nada AlMohaisen.

**Visualization:** Nada AlMohaisen.

**Writing – original draft:** Nada AlMohaisen.

**Writing – review & editing:** Nada AlMohaisen, Sorrel Burden.

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
