## [Decision Letter · Decision Letter 0]

28 Apr 2022

PONE-D-21-29906

What is the overlap between malnutrition, frailty and sarcopenia in the older population?  Study protocol for cross-sectional study using UK Biobank

PLOS ONE

Dear Dr. AlMohaisen,

Thank you for submitting your manuscript to PLOS ONE. After careful consideration, we feel that it has merit but does not fully meet PLOS ONE’s publication criteria as it currently stands. Therefore, we invite you to submit a revised version of the manuscript that addresses the points raised during the review process.

We look forward to receiving your revised manuscript.

Kind regards,

Melissa M Markofski

Academic Editor

PLOS ONE

“Yes, this PhD is scholarship from the Saudi Arabia cultural bureau in London, which they pay the tutors without interfering with the works.”

“NO authors have competing interests”

4. We note that you have stated that you will provide repository information for your data at acceptance. Should your manuscript be accepted for publication, we will hold it until you provide the relevant accession numbers or DOIs necessary to access your data. If you wish to make changes to your Data Availability statement, please describe these changes in your cover letter and we will update your Data Availability statement to reflect the information you provide

5. Please note that in order to use the direct billing option the corresponding author must be affiliated with the chosen institute. Please either amend your manuscript to change the affiliation or corresponding author, or email us at plosone@plos.org with a request to remove this option.

6. PLOS requires an ORCID iD for the corresponding author in Editorial Manager on papers submitted after December 6th, 2016. Please ensure that you have an ORCID iD and that it is validated in Editorial Manager. To do this, go to ‘Update my Information’ (in the upper left-hand corner of the main menu), and click on the Fetch/Validate link next to the ORCID field. This will take you to the ORCID site and allow you to create a new iD or authenticate a pre-existing iD in Editorial Manager. Please see the following video for instructions on linking an ORCID iD to your Editorial Manager account: https://www.youtube.com/watch?v=_xcclfuvtxQ.

Reviewers' comments:

Reviewer's Responses to Questions

**Comments to the Author**

1. Does the manuscript provide a valid rationale for the proposed study, with clearly identified and justified research questions?

Reviewer #1: Yes

Reviewer #2: Yes

2. Is the protocol technically sound and planned in a manner that will lead to a meaningful outcome and allow testing the stated hypotheses?

Reviewer #1: Partly

Reviewer #2: Yes

3. Is the methodology feasible and described in sufficient detail to allow the work to be replicable?

Reviewer #1: Yes

Reviewer #2: Yes

4. Have the authors described where all data underlying the findings will be made available when the study is complete?

Reviewer #1: Yes

Reviewer #2: Yes

5. Is the manuscript presented in an intelligible fashion and written in standard English?

Reviewer #1: Yes

Reviewer #2: No

6. Review Comments to the Author

You may also provide optional suggestions and comments to authors that they might find helpful in planning their study.

Reviewer #1: In this submission, the authors propose a cross-sectional study to determine the prevalence and overlap between the related conditions of malnutrition, frailty, and sarcopenia. They will do so utilizing a subset of data from the UK Biobank database. Specifically, they will utilize data from adults 50 to 69 years old. A clear strength of the proposed study is the anticipated subject number of more than 380,000 subjects. In general, the paper was very well written. I do have a few concerns/suggestions for the authors.

1. One of the three phenotypic criteria for malnutrition indicates a cut-point for BMI of < 22 kg/m2 for adults older than 70. However, this data set will not contain subjects over 70.

2. One of the etiologic criteria for malnutrition indicates that CRP and albumin will be used as indices of inflammation. Albumin is not a marker of inflammation, but could be an indicator of malnutrition. I can see including albumin as a marker of malnutrition, but the authors should not indicate that it is an indicator of inflammation. I’m referring specifically to the wording the in second paragraph of page 7.

3. The authors indicate that they are using two models to map frailty—the 36 deficits model and the phenotype model. Under the description of each (pages 7 and 8) they define the criteria for frailty, but it’s less clear how the subjects will be ultimately classified. To be considered frail, do subjects need to meet the criteria under both models (or just one)?

4. Also, the supplementary table for the 36 deficits models is not well labeled. In the text it refers to Table S1. The table is not labeled. It is also not clear in the text that the BioBank does not include 3 of the 36 deficits.

5. For sarcopenia, the authors are using the EWGSOP definition. While they will have measurements of muscle mass (BIA) and muscle strength (handgrip), they must reply on self-reported data for physical function. The criterion for physical function in the EWGSOP is gait speed. Under the section “mapping sarcopenia” the authors state that data from IPAQ will be used to assess physical function. Neither the short nor the long form of the IPAQ ask questions regarding gait speed. Rather they return an estimate of habitual PA in METxmin/wk. Thus, the wording in this section is incorrect. Did the authors mean to state that they will use the responses to the walking speed question in table 3? If so, are their studies suggesting that self-reported, “slow pace” walking speed is significantly correlated with the cut-points for gait speed suggested in the EWGSOP definition? I know that the authors are constrained by the data collected in the UK BioBank, but they need to acknowledge that relying on a self-reported assessment of walking speed is a limitation in diagnosing sarcopenia in this population. This is somewhat addressed in the discussion section, but not specifically.

Reviewer #2: The background gives a good overview of the topic. Methods/design are very well described, and the variables are clearly illustrated in table 1, 2. The statistical analyses that will be used are satisfactory presented. Discussion and study limitations are conclusive stating “using a secondary database not designed specifically to answer the research question resulting in selection bias in the sample". The authors also highlight the limitations because some of the variables for identification of malnutrition, frailty or sarcopenia will not be available or are self-reported.

Language check!

Minor comments; Abstract: Discussion: This proposed study will help with understanding the presence (I suggest: clarify or help to understand).

Background, page 3, 2nd paragraph, first sentence “alterations to the senses” do you mean “in”?

2nd paragraph line 3-6, rewrite. I suggest adding ref 19 (fig 3 a conceptual tree of nutritional disorders) along with ref 9 and focus on malnutrition i.e.: a) starvation derived undernutrition and b) cachexia – disease related malnutrition! It is important to include “disease-related malnutrition” in the text.

Page 7, line 2 some word (s) is missing here!

Page 9, paragraph 4 second line – word (s) missing – will be collected?

Discussion, line 3. Here you may add a study performed in nursing home setting: Line 7-8 word (s) missing!

Faxén-Irving G, Luiking Y, Grönstedt H, Franzén E, Seiger Å, Vikström S, Wimo A, Boström AM, Cederholm T. Do Malnutrition, Sarcopenia and Frailty Overlap in Nursing-Home Residents? https://doi.org/10.14283/jfa.2020.45 J frailty Aging 2021;10(1)17-21.

7. PLOS authors have the option to publish the peer review history of their article (what does this mean?). If published, this will include your full peer review and any attached files.

Reviewer #1: No

Reviewer #2: No

---

## [Author Response · Author response to Decision Letter 0]

6 Jun 2022

The authors would like to thank the reviewers for their insightful comments that have improved the quality of our manuscript. All comments have been addressed as shown in the word document (Reviewer Respond)

---

## [Decision Letter · Decision Letter 1]

22 Sep 2022

PONE-D-21-29906R1What is the overlap between malnutrition, frailty and sarcopenia in the older population?  Study protocol for cross-sectional study using UK BiobankPLOS ONE

Dear Dr. AlMohaisen,

Thank you for submitting your manuscript to PLOS ONE. After careful consideration, we feel that it has merit but does not fully meet PLOS ONE’s publication criteria as it currently stands. Therefore, we invite you to submit a revised version of the manuscript that addresses the points raised during the review process.

We look forward to receiving your revised manuscript.

Kind regards,

Sultana Monira Hussain

Academic Editor

PLOS ONE

Additional Editor Comments:

I agree with several of the comments raised by the reviewers. Please, consider reviewing all those issues raised.

Reviewers' comments:

Reviewer's Responses to Questions

**Comments to the Author**

1. Does the manuscript provide a valid rationale for the proposed study, with clearly identified and justified research questions?

Reviewer #2: Yes

Reviewer #3: Yes

2. Is the protocol technically sound and planned in a manner that will lead to a meaningful outcome and allow testing the stated hypotheses?

Reviewer #2: Yes

Reviewer #3: Yes

3. Is the methodology feasible and described in sufficient detail to allow the work to be replicable?

Reviewer #2: Yes

Reviewer #3: Yes

4. Have the authors described where all data underlying the findings will be made available when the study is complete?

Reviewer #2: Yes

Reviewer #3: Yes

5. Is the manuscript presented in an intelligible fashion and written in standard English?

Reviewer #2: Yes

Reviewer #3: Yes

6. Review Comments to the Author

You may also provide optional suggestions and comments to authors that they might find helpful in planning their study.

Reviewer #2: The MS has improved!

Some minor comments, most language:

Page 3, line 55-58: “Malnutrition has two forms: undernutrition, which involves wasting, stunted growth (starvation related under-weight and cachexia or disease related malnutrition), vitamin or mineral deficiencies (micronutrient abnormalities), and being overweight or obese (overnutrition)”.

My suggestion: There are (or exist) two forms of malnutrition…….

Table 2. Variable mapped to malnutrition, frailty and sarcopenia. This table is difficult to read – it’s some kind of overlapping in the pdf-file.

Page 10, Line 202-204: word is missing. “These variables will represent three phenotypic criteria and two etiologic criteria, where patients will be defined as having malnutrition if …..has one phenotypic criterion and one aetiological criterion.

Page 13, Line 278-280: words are missing: “From the baseline assessment, details of the participants’ characteristics anthropometric measurements, participants’ medical histories, smoking status, uses of medications, dietary intake records and alcohol intake”……..

Discussion. Page 14, Line 288-292: A study measured the overlap between the three conditions in a nursing home, which showed that 7% of the 92 residents aged 75 and above had all three conditions, and 32% of the 290 participants were free from malnutrition, frailty, and sarcopenia. In addition, 5% of participants were malnourished with sarcopenia in the nursing home. Whereas 10% patients had sarcopenia and were frail 58].

My suggestion: In a group of 92 selected nursing home residents aged 75 and above all three conditions were present in 7%. Twenty-nine (32%) residents were neither malnourished, sarcopenic nor frail. In addition, 5% of the residents were malnourished and simultaneously sarcopenic and frailty coexisted with sarcopenia in 10% of the subjects.

Line 296-298. word is missing: “This highlights …….early identification of these conditions in older adults is vital so treatment and interventions can initiated in this age group”

Reviewer #3: Authors reports that baseline assessment will be used for the study. Therefore, since participants aged between 40-69 were included, I do not expect to have participants older than 70 years in analyses. This point is unclear and deserved better explanation.

It is not clear why authors expect to include 76% of the total sample of the UK biobank in the analyses. Please develop.

Table 1 and Table 2 are unreadable in the PDF document I have received. Therefore I was unable to review the criteria of sarcopenia, frailty and malnutrition, which is unfortunate.

I recommend authors to use the EWGSOP2 criteria for the diagnosis of sarcopenia. Please update criteria and cut-offs used for the diagnosis.

Could author inform if tools used for the measures of muscle mass and muscle strength were calibrated ? This is an important aspect.

7. PLOS authors have the option to publish the peer review history of their article (what does this mean?). If published, this will include your full peer review and any attached files.

Reviewer #2: No

Reviewer #3: No

---

## [Author Response · Author response to Decision Letter 1]

11 Oct 2022

Thank you for your time and effort. Please find, all your valuable comments have been address in separate word document, attached as (Rebuttal 10 Oct 2022)

---

## [Decision Letter · Decision Letter 2]

16 Nov 2022

What is the overlap between malnutrition, frailty and sarcopenia in the older population?  Study protocol for cross-sectional study using UK Biobank

PONE-D-21-29906R2

Dear Dr. AlMohaisen,

We’re pleased to inform you that your manuscript has been judged scientifically suitable for publication and will be formally accepted for publication once it meets all outstanding technical requirements.

Kind regards,

Sultana Monira Hussain

Academic Editor

PLOS ONE

Additional Editor Comments (optional):

Reviewers' comments:

Reviewer's Responses to Questions

**Comments to the Author**

1. Does the manuscript provide a valid rationale for the proposed study, with clearly identified and justified research questions?

Reviewer #2: Yes

Reviewer #3: Yes

2. Is the protocol technically sound and planned in a manner that will lead to a meaningful outcome and allow testing the stated hypotheses?

Reviewer #2: Yes

Reviewer #3: Yes

3. Is the methodology feasible and described in sufficient detail to allow the work to be replicable?

Reviewer #2: Yes

Reviewer #3: Yes

4. Have the authors described where all data underlying the findings will be made available when the study is complete?

Reviewer #2: Yes

Reviewer #3: No

5. Is the manuscript presented in an intelligible fashion and written in standard English?

Reviewer #2: Yes

Reviewer #3: Yes

6. Review Comments to the Author

You may also provide optional suggestions and comments to authors that they might find helpful in planning their study.

Reviewer #2: The MS has improved and the study will be interesting to read about later on!

Some minor (language) comments:

Page 3, Line 55-56: There are two forms of malnutrition undernutrition, and being overweight or obese which results from diet-related non-communicable diseases [9, 10].

Suggest: There are two forms of malnutrition: undernutrition and being……..

Page 7, line 151: Then a series of assessment stations where undertaken….

should be were not where

Reviewer #3: Authors responded well to my previous comments. I am favorable with the publication of this protocol.

7. PLOS authors have the option to publish the peer review history of their article (what does this mean?). If published, this will include your full peer review and any attached files.

Reviewer #2: No

Reviewer #3: **Yes: **Charlotte Beaudart

---

## [Editor Report · Acceptance letter]

25 Nov 2022

PONE-D-21-29906R2 

What is the overlap between malnutrition, frailty and sarcopenia in the older population? Study protocol for cross-sectional study using UK Biobank 

Dear Dr. AlMohaisen:

I'm pleased to inform you that your manuscript has been deemed suitable for publication in PLOS ONE. Congratulations! Your manuscript is now with our production department. 

Kind regards, 

on behalf of

Dr. Sultana Monira Hussain 

Academic Editor

PLOS ONE